# Rare among Rare: Phenotypes of Uncommon CMT Genotypes

**DOI:** 10.3390/brainsci11121616

**Published:** 2021-12-08

**Authors:** Luca Gentile, Massimo Russo, Federica Taioli, Moreno Ferrarini, M’Hammed Aguennouz, Carmelo Rodolico, Antonio Toscano, Gian Maria Fabrizi, Anna Mazzeo

**Affiliations:** 1Department of Clinical and Experimental Medicine, University of Messina, 98125 Messina, Italy; russom@unime.it (M.R.); aguenoz.mhommed@unime.it (M.A.); crodolico@unime.it (C.R.); antonio.toscano@unime.it (A.T.); annamazzeo@yahoo.it (A.M.); 2Department of Neurological Sciences, Biomedicine and Movement Sciences, University of Verona, Piazzale L.A. Scuro 10, 37134 Verona, Italy; federica.taioli@univr.it (F.T.); moreno.ferrarini@univr.it (M.F.); gianmaria.fabrizi@univr.it (G.M.F.); 3Azienda Ospedaliera Universitaria Integrata Verona—Borgo Roma, Piazzale L.A. Scuro 10, 37134 Verona, Italy

**Keywords:** CMT, rare genes, genotype/phenotype

## Abstract

(1) Background: Charcot–Marie–Tooth disease (CMT) is the most frequent form of inherited chronic motor and sensory polyneuropathy. Over 100 CMT causative genes have been identified. Previous reports found *PMP22*, *GJB1*, *MPZ*, and *MFN2* as the most frequently involved genes. Other genes, such as *BSCL2*, *MORC2*, *HINT1*, *LITAF*, *GARS*, and autosomal dominant *GDAP1* are responsible for only a minority of CMT cases. (2) Methods: we present here our records of CMT patients harboring a mutation in one of these rare genes (*BSCL2*, *MORC2*, *HINT1*, *LITAF*, *GARS*, autosomal dominant *GDAP1*). We studied 17 patients from 8 unrelated families. All subjects underwent neurologic evaluation and genetic testing by next-generation sequencing on an Ion Torrent PGM (Thermo Fischer) with a 44-gene custom panel. (3) Results: the following variants were found: *BSCL2* c.263A > G p.Asn88Ser (eight subjects), *MORC2* c.1503A > T p.Gln501His (one subject), *HINT1* c.110G > C p.Arg37Pro (one subject), *LITAF* c.404C > G p.Pro135Arg (two subjects), *GARS* c.1660G > A p.Asp554Asn (three subjects), *GDAP1* c.374G > A p.Arg125Gln (two subjects). (4) Expanding the spectrum of CMT phenotypes is of high relevance, especially for less common variants that have a higher risk of remaining undiagnosed. The necessity of reaching a genetic definition for most patients is great, potentially making them eligible for future experimentations.

## 1. Introduction

Charcot–Marie–Tooth disease (CMT) is the most frequent form of inherited chronic motor and sensory polyneuropathies and one of the most frequent genetic neuromuscular disorders, with a prevalence of 1:2500 [1]. CMT can manifest in heterogeneous ways, with variable phenotypic presentation even among subjects belonging to the same family [2]. In rare cases, the CMT phenotype could be worsened by a superimposed inflammatory process [3,4]. The classification of CMT is based on the type of inheritance (autosomal dominant, AD; autosomal recessive, AR; X-linked) and on the results of upper limb motor nerve conduction studies: CMT1, predominantly demyelinating, is characterized by nerve velocity under 38 m/s; CMT2, predominantly axonal, presents motor velocities above 38 m/s. An additional group includes intermediate CMT, with motor conduction velocities between 25 and 45 m/s [5]. Over 100 CMT causative genes have been identified. Previous reports found *PMP22*, *GJB1*, *MPZ*, and *MFN2* as the most frequently involved genes [6,7,8,9]. Other reports of CMT patients from the Mediterranean area showed that other genes (autosomal recessive *GDAP1* and *HSPB1*) could frequently be involved, suggesting that genetic distribution could possibly be influenced by geographical features [10,11,12,13,14]. It has been calculated that almost 90% of CMT patients harbor a mutation in one of the above-mentioned genes [1,15]. Other genes, such as *BSCL2*, *MORC2*, *HINT1*, *LITAF*, *GARS1*, and autosomal dominant *GDAP1* are responsible of only a minority of CMT cases, with the consequence that the corresponding phenotypic presentations are less known and less defined.

*BSCL2*, encoding for Seipin, an integral membrane protein, has been associated with an autosomal dominant transmitted form of distal hereditary motor neuropathy (dHMNtype V) and axonal CMT (CMT2D) (Table 1) and with Silver syndrome (SS)/spastic paraplegia 17 [16]. It is also involved in autosomal recessive congenital generalized lipodystrophy type 2 [16]. The *BSCL2* pathogenic variants most frequently reported are p.Asn88Ser and p.Ser90Leu, while p.Ser90Trp and p.Arg96His are less frequent [17]. Recently, two other disease-causative variants have been identified (p.Asn88Thr and p.Ser141Ala) [17]. *BSCL2* patients have been reported worldwide [17,18,19,20,21,22,23,24,25]. Disease onset is between the 2nd and the 5th decade, with apparently a greater disease severity in male patients than in females [17]. Clinical examination could reveal distal motor weakness in the upper and/or lower limbs and pyramidal signs [24,25]. Sensory involvement could be present, generally of mild degree and often subclinical, and can be detected only at neurophysiological examination [16,17,18,21,22,25]. Sporadic cases of respiratory disfunction and sensory hearing loss have been reported [18,22] (Table 1).

A *MORC2* variant associated with CMT (Table 1) has been firstly described in 2015 [26]. This gene codes for Microrchidia family CW-type zinc finger 2 (MORC2), a member of the MORC protein family, that has been postulated to regulate the DNA damage response [26]. In a recent report, *MORC2* variants have been found responsible for the modification of axonal transport, neurofilament homeostasis, and the architecture of the cytoskeleton; they possibly have a role also in Schwann cell function [27]. The most frequent pathogenic variant is p.Arg252Trp, which is presents in more than 50% of cases [27]. The phenotypic manifestation correlated with this gene variant is an axonal motor and sensory polyneuropathy (CMT2Z) with an onset ranging from the congenital period, associated with a more severe clinical picture resembling Spinal Muscular Atrophy, to the juvenile age, characterized by classic distal sensory and motor symptoms and signs of disease progressively and asymmetrically spreading to proximal sites (Table 1). Pyramidal signs, cerebellar atrophy, diaphragmatic paralysis, tongue atrophy, mental retardation, and spinal cord atrophy were also reported [28] (Table 1).

*HINT1* (Table 1) encodes histidine triad nucleotide-binding protein 1 (*HINT1*) that is involved in manifold transcriptional and signaling pathways [29]. Autosomal recessive variants of this gene have been previously reported, in particular from Central and South-East Europe, Russia, and Turkey. Four proven founder variants have been identified: p.Arg37Pro, the most common, p.Cys84Arg, p.His112Asn, and Cys38Arg [30,31]. Disease onset is frequently in the first or second decade, with a prevalent motor polyneuropathy that causes weakness in lower limbs’ muscles and gait impairment [32]. In most of the patients, neuromyotonia is also present [31,32] (Table 1). Recently, even neuro-psychiatric symptoms (late language development, social behavioral alterations) have been described [31,33] (Table 1).

Lipopolysaccharide-induced tumor necrosis factor (LITAF) is involved in an endolysosomal pathway that is required to maintain the homeostasis of late endosomes and lysosomes [34]. *LITAF* pathogenic variants cause an autosomal dominant chronic demyelinating polyneuropathy, classified as CMT1C [35] (Table 1), frequently but not exclusively presenting in young age [36]. The CMT1C phenotype could resemble that of CMT1A, in particular as regard the degree of sensory loss, foot deformities, and scoliosis, but muscle weakness, areflexia, and motor nerve conduction are milder [37]. Disease presentation with only sensory symptoms (limited to isolated paresthesia) could also be possible [37], as well as the presence of motor conduction blocks, postural tremors, and plantar ulcers [35,37] (Table 1).

*GARS1* encodes Glycyl-tRNA synthetase, a dually (cytoplasmic and mitochondrial) localized enzyme of the aminoacyl-tRNA synthetase (aaRS) family [38,39]. Defects in this enzyme’s activity result in the reduction of aminoacylation activity, changes in axon location, and alterations in the neuropilin 1 pathway [38] and lead to a CMT2 or dHMN type V phenotype, mainly involving the upper extremities at distal sites [16] (Table 1).

Ganglioside-induced differentiation-associated protein 1 (*GDAP1*) is an integral mitochondrial membrane protein. It is mainly expressed in neurons of both central and peripheral nervous systems, and has the role of maintaining the normal functioning, structure, and movement of the mitochondria [40]. Autosomal recessive variants of *GDAP1* have been associated with demyelinating, intermediate, and axonal CMT [41], whereas the few cases of dominant variants are mainly responsible for axonal CMT [42,43] (Table 1).

We present here our records of CMT patients harboring a mutation in one of these rare genes (*BSCL2*, *MORC2*, *HINT1*, *LITAF*, *GARS1*, autosomal dominant *GDAP1*).

## 2. Materials and Methods

We studied 17 patients from 8 unrelated family. All subjects underwent a neurologic evaluation and a genetic test by next-generation sequencing on an Ion Torrent PGM (Thermo Fischer, Waltham, MA, USA) with a 44-gene custom panel (see Appendix A), and we found *BSCL2*, *MORC2*, *HINT1*, *LITAF*, *GARS1*, or *GDAP1* variants. Nerve conduction studies were performed in 12/17 patients. The pathogenicity of the found variants was investigated through the VarSome platform [44]. This study was approved and performed under the ethical guidelines issued by our institution for clinical studies (ethical Committee of the University Hospital of Messina; address: AOU “G.Martino,” via Consolare Valeria n. 1, 98125-Messina (ME), Italy. Approval Code: prot. 53, verbale n. 03/2014; approval date: 24 February 2014) and was in compliance with the Helsinki Declaration. The diagnostic procedures were conducted according to the ethics committee of our hospital, and informed written consent was obtained from all patients.

## 3. Results

### 3.1. BSCL2 (CMT2) (OMIM 606158)

We report here two unrelated three-generation families harboring the c.263A > G p.Asn88Ser heterozygous pathogenic variant of *BSCL2* (Table 2 and Table 3).

In the first family (Figure A1), the proband (patient 1.1) presented distal motor weakness in the hands, with brisk reflexes and no sensory disturbances. However, neurophysiological study revealed a predominantly motor axonal neuropathy (Table 4); therefore, a subclinical sensitive involvement could be supposed. A neurological examination of one of her sisters (patient 1.2) revealed with pes cavus, distal motor deficits in the lower limbs, hypoesthesia and dysesthesias of the feet, and pyramidal signs (hyperreflexia and bilateral Hoffman’s sign). Another sister (patient 1.3) had brisk reflexes and bilateral Hoffman’s sign; since the age of 6, she presented visual impairment and was diagnosed with pigmentosus retinitis and bilateral cataract. The proband’s son (patient 1.4), 16 years old, showed normal features at neurological and neurophysiological examinations, except for bilateral sensory hypoacusia. The proband’s father (patient 1.5), 74 years of age, had only a bilateral pes cavus. Patients 1.2, 1.3, and 1.4 had an axonal polyneuropathy at neurophysiologic evaluation (Table 4).

In the second family (Figure A2), the proband (patient 2.1) was a 16-year-old boy who had presented pes cavus and hammertoes for many years. He mainly complained of cramps at his lower limbs. At neurological examination, we found bilateral pes cavus and hammertoes, with the left foot intrarotated, bilateral shortening of Achille’s tendon, proximal and distal muscular hypotrophy, and hyposthenia of the upper limbs, hyposthenia of plantar dorsiflexion, hyperreflexia of the lower limbs, and no sensory impairment. The proband’s father (patient 2.2) was 43 years old and did not complain of any symptoms, although he referred that many people had told him that his walking was not normal. At the visit, we found bilateral pes cavus with hammertoes and hypotrophy and hyposthenia of the antero-lateral legs’ muscles with pseudohypertrophy of the calves’ muscles. The patellar reflex was brisk bilaterally, but Achilles’ tendon reflex was normal. Sensory examination was normal. Neurophysiological studies revealed an axonal polyneuropathy (Table 4) and an augmentation of the central conduction time in upper limbs motor evoked potentials in both patients. The proband’s grandmother (patient 2.3), 72 years old, had only bilateral pes cavus and hammertoes.

### 3.2. MORC2 (CMT2) (OMIM 616661)

Our patient (patient 3.1, Table 3), a 54-year-old female, had a three-year story of paresthesia in both feet and hands and impaired walking and balance. Admitted to another Hospital, she underwent liquor analysis that revealed the presence of 114 mg/dL of protein. A diagnosis of CIDP was made, and she was treated with intravenous immunoglobulins, without benefit. When she came to our observation, a neurologic examination showed ataxic gait, deep tendon reflexes diminished in the upper limbs and absent in the lower limbs, and decreased pain and vibration sensitivity distally in the four limbs, with dysesthesias. Nerve conduction studies showed an axonal demyelinating polyneuropathy (Table 4). Genetic testing showed the presence of a c.1503A > T p.Gln501His heterozygous variant of uncertain significance (VUS) in *MORC2* (Table 2).

### 3.3. HINT1 (CMT2) (OMIM 601314)

Our patient (patient 4.1, Table 3), a 15-year-old female of Eastern Europe origin, came to our observation with a likely positive story of neuromuscular disorder: a cousin of her mother, dead at age 34, had motor deficits in the four limbs and reduced visual acuity. The patient’s parents were not consanguineous. She had complained of gait impairment since the age of 10. Some years after, she developed difficulty in fine hands movements and balance impairment. Neurologic examination showed bilateral stepping gait, hypotrophy of hands’ and legs’ muscles, shortening of Achille’s tendons, postural tremor in both hands, moderate/severe hyposthenia of intrinsic hand muscles and of antero-lateral leg’s compartment muscles, areflexia. Neurophysiologic evaluation showed an axonal-demyelinating sensory–motor polyneuropathy, more severe in the lower limbs (Table 4). Genetic tests revealed a c.110G > C p.Arg37Pro homozygous, pathogenic variant in *HINT1* (Table 2).

### 3.4. LITAF (CMT1) (OMIM 603795)

Our proband (patient 5.1, Table 3) (Figure A3), a 41-year-old male, presented pes cavus since adolescence. However, he did not report any kind of symptoms until the age of 37, when he developed foot paresthesia, walking difficulty, and reduced right grip strength. He underwent a neurophysiological examination that showed demyelination patterns consisting with CMT1 (Table 4). At neurological examination, he showed distal upper limbs postural tremors, impossibility of walking on heels, positive Romberg test, hyposthenia of intrinsic hand muscles (MRC 4) and of leg (MRC 4−) and calf muscles (MRC 4+), deep tendon areflexia, hypoesthesia, and hypopallesthesia of the feet. The proband’s daughter (patient 5.2), 6 years old, had complained of walking difficulties (easy falls, walking on toes) since the age of 2. A genetic test of both patients revealed the presence of the c.404C > G p.Pro135Arg heterozygous, uncertain-significance/likely pathogenic variant of *LITAF* (Table 2).

### 3.5. GARS1 (CMT2) (OMIM 600287)

We report here a two-generation family with the c.1660G > A p.Asp554Asn heterozygous, uncertain-significance/likely pathogenic variant on *GARS1* (Table 2 and Table 3) (Figure A4).

The proband (patient 6.1) is a 53-year-old male that reported no family history suggestive of neuromuscular disease. For a couple of years, he had been complaining of a no better defined “pain” in the lower limbs and of a reduction of hands’ strength (i.e., he had become unable to unscrew a bottle cap). At neurological examination, he could not stay on heels and presented hypotrophy of tenar and leg muscles, with mild hyposthenia of antero-lateral leg muscles. Deep tendon reflexes were reduced/absent in the upper limbs and normal in the lower limbs. Sensory examination was normal. A neurophysiological evaluation showed an axonal polyneuropathy (Table 4). Genetic testing revealed the presence of a c.1660G > A p.Asp554Asn heterozygous variant of *GARS*. After genetic counselling, both proband’s sons (14 and 28 years old) (patients 6.2 and 6.3) underwent genetic testing that revealed the same variant in both: their neurological examination was normal, but the younger one complained of mild difficulty in fine hands’ movements and sporadic intentional tremor.

### 3.6. GDAP1 (CMT1/CMT2) (OMIM 606598)

We present two unrelated patients, both harboring the c.374G > A p.Arg125Gln heterozygous, likely pathogenic variant of *GDAP1* (Table 2 and Table 3).

The first patient (patient 7.1), a 52-year-old male, referred of a walking impairment since the age of 43, that gradually worsened during the years. He did not report any family history. Clinical evaluation showed bilateral steppage, severe lower limbs’ muscles hypotrophy, severe hyposthenia of the leg muscles (MRC 2), and mild hyposthenia of the calf muscles (MRC 4). He presented also hyposthenia of the proximal lower limbs’ muscles and was able to squat only with support. Deep tendon reflexes were very brisk, except for the Achille’s tendon reflexes (absent).

The second patient (patient 8.1), a 50-year-old male, presented, since the age of 3, with a reduction of visual acuity. At the age of 20, he was diagnosed with optic nerve subatrophy and neurosensorial hypoacusia. At the age of 24, he developed a sensory–motor polyneuropathy. A neurological evaluation showed left steppage, reduction of lower limbs muscles’ tonus and trophism, mild hyposthenia of the anterolateral leg muscles (worse on the left side), and brisk reflexes in the upper limbs, with bilateral Hoffman’s sign and hypopallestesia from the anterior superior iliac spine.

Nerve conduction studies revealed an axonal polyneuropathy in both patients (Table 4).

## 4. Discussion

CMT genetic confirmation has always represented a challenge. Classically, clinicians could use some algorithms, based on NCV and type of inheritance, to guide the genetic testing [15]. In this way, approximately 60% of CMT patients could receive a genetic diagnosis [1]. Recently introduced innovative techniques of genome sequencing, such as next-generation sequencing (NGS) and whole-exome sequencing (WES), have rapidly increased the number of known genes associated with CMT, which are now over 100 [5]. Although some genes are more frequently found than others [6,7,8,9,10,11,12,13,14], also ‘rare among rare’ genes should be taken into consideration. In this paper, we have presented some patients harboring genetic variants of uncommon CMT genes. Almost all of them (*BSCL2*, *MORC2*, *HINT1*, *GARS1*, and autosomal dominant *GDAP1*) are responsible for a CMT2 phenotype, types of CMT for which genetic definition is less frequently reached. Some peculiar features that could be associated with these variants could help clinicians direct the genetic testing. For example, in our *BSCL2* patients, we found the already reported pyramidal signs and neurosensorial hypoacusia. In addition, we reported pigmentosus retinitis with bilateral cataract in patient 1.3 and proximal UL weakness in pt. 2.1. Pigmentosus retinitis has been associated with PHARC syndrome [46] and with *ATP6* [47] and *MTMR2* gene variants [48]. Recently, a dysfunction of the endoplasmic reticulum membrane protein complex (EMC), a conserved player in the process of membrane protein biogenesis, has been reported to play a role in the pathogenesis of certain congenital diseases such as cystic fibrosis, pigmentosus retinitis, and Charcot–Marie–Tooth disease [49]. The hypothesis is that these pathologies are caused by mutations within membrane proteins that require the EMC during their production [49]. The *MORC2* VUS patient had no particular abnormal characteristics, with the exception of an increase in CSF protein level that initially led to a CIDP misdiagnosis. As this gene is considered to play a role in the development of the nervous system [28], it could be supposed that it may cause an alteration of the blood–brain barrier that could justify this finding. Moreover, the absence of temporal dispersion or conduction blocks in nerve conduction studies, the slowly progressive clinical picture, and the loss of improvement after adequate therapy (intravenous immunoglobulin) made the CIDP diagnosis very improbable. In the *LITAF* and *GARS1* patients, we confirmed the presence of foot deformities in the first and a prominent distal upper limb involvement in the second. Peculiar was the phenotype of the two patients with the *GDAP1* variant, who presented with proximal/asymmetric lower limb weakness, clinical signs of first motoneuron involvement, and optic and acoustic nerve disorders.

## 5. Conclusions

Expanding the spectrum of CMT phenotypes is of high relevance, especially as it allows identifying less common mutations that have a higher risk of remaining undiagnosed. A better knowledge of the clinical manifestation of these rare mutations could be crucial to correctly address genetic testing, in particular considering that NGS and WES are still not widespread, especially in some health systems. Moreover, now that a phase III clinical trial (PLEO-CMT) is available for CMT patients with a specific mutation (*PMP22* duplication, CMT1A) [50], the necessity of reaching a genetic definition for most of the patients is even greater, as it may make them eligible for future experimentations.

## Figures and Tables

**Table 1 brainsci-11-01616-t001:** Usual phenotypes and atypical features of the *BSCL2, MORC2, HINT1, LITAF, GARS1*, and *GDAP1* genes.

Gene	Phenotypes	Atypical Clinical Features
*BSCL2*	AD-CMT2	pyramidal signs [18,22,24,25] respiratory dysfunction [18] sensory hearing loss [22]
*MORC2*	AD-CMT2	asymmetric proximal weakness [28] pyramidal signs [28] cerebellar atrophy [28] diaphragmatic paralysis [28] tongue atrophy [28] mental retardation [28] spinal cord atrophy [28]
*HINT1*	AD-CMT2	Neuromyotonia [31,32] neuro-psychiatric symptoms [32,33]
*LITAF*	AD-CMT1	young age of onset [36] foot deformities [35,37] plantar ulcers [35,37] scoliosis [35,37]
*GARS1*	AD-CMT2	prominent distal upper limb involvement [16]
*GDAP1*	AR-CMT1/AD-CMT2	early and rapidly progressive phenotype (AR-CMT1) [41]

**Table 2 brainsci-11-01616-t002:** Genetic analysis results of the eight families studied.

Family	Gene Reference Transcript	Variant (cDNA Protein)	Allele ID	Genotype	Classification (ACMG 2015 Guidelines)	ACMG/AMP Criteria Codes	Variant First Reported by:
1, 2	*BSCL2* NM_001130702	c.263A > G p.Asn88Ser	931315	htz	pathogenic	PS3, PM2, PP3, PP5	[19]
3	*MORC2* NM_014941.3	c.1503A > T p.Gln501His	n.a.	htz	Uncertain significance	PM2	Present report
4	*HINT1* NM_005340.6	c.110G > C p.Arg37Pro	45887	hmz	pathogenic	PVS1, PS3, PM1, PM2, PP2, PP3, PP5	[45]
5	*LITAF* NM_001136472.1	c.404C > G p.Pro135Arg	204476	htz	Uncertain significance/likely pathogenic	PM2, PP3	[35]
6	*GARS1* NM_002047	c.1660G > A p.Asp554Asn	24247	htz	Uncertain significance/likely pathogenic	PS3	[39]
7, 8	*GDAP1* NM_018972	c.374G > A p.Arg125Gln	315143	htz	Likely pathogenic	PM1, PM2, PP2, PP3	Present report

htz, heterozygous; hmz, homozygous; n.a.: not available. ACGM/AMP codes: PVS1: Null variant (nonsense, frameshift, canonical ± one or two splice sites, initiation codon, single or multiexon deletion) in a gene where LOF is a known mechanism of disease (Pathogenic, Very Strong); PS3: Well-established in in vitro or in vivo functional studies supportive of a damaging effect on the gene or gene product (Pathogenic, Strong); PM1: Located in a mutational hot spot and/or in a critical and well-established functional domain (e.g., active site of an enzyme) without benign variation (Pathogenic, Moderate); PM2: Absent from controls (or at extremely low frequency if recessive) in Exome Sequencing Project, 1000 Genomes Project, or Exome Aggregation Consortium (Pathogenic, Moderate); PP2: Missense variant in a gene that has a low rate of benign missense variation and in which missense variants are a common mechanism of disease (Pathogenic, Supporting); PP3: Multiple lines of computational evidence support a deleterious effect on the gene or gene product (conservation, evolutionary, splicing impact, etc.) (Pathogenic, Supporting); PP5: Reputable source recently reports the variant as pathogenic, but the evidence is not available to the laboratory to perform an independent evaluation (Pathogenic, Supporting). Allele ID was obtained from ClinVar.

**Table 3 brainsci-11-01616-t003:** Clinical characteristics and unusual features of all of the 17 subjects.

Family	Gene	Variant	Patient	Age (yrs)	Weakness		Sensory Deficit		Reflexes		Unusual Features
					UL	LL	UL	LL	UL	LL	
1	*BSCL2*	c.263A > G p.Asn88Ser	1.1 (p)	39	++	/	/	/	+++	++	Pyramidal signs
			1.2	43	/	+	/	++	++	++	Pyramidal signs
			1.3	48	+	+	/	/	/	+++	pigmentosus retinitis, bilateral cataract, pyramidal signs
			1.4	16	/	/	/	/	/	/	bilateral sensory hypoacusia
			1.5	74	/	/	/	/	/	/	/
2	*BSCL2*	c.263A > G p.Asn88Ser	2.1 (p)	16	+++	+	/	/	/	++	Proximal UL weakness, pyramidal signs
			2.2	43	/	+	/	/	/	+	N.A.
			2.3	72	/	/	/	/	/	/	N.A.
3	*MORC2*	c.1503A > T p.Gln501His	3.1	54	/	/	++	++	--	---	High CSF protein
4	*HINT1*	c.110G > C p.Arg37Pro	4.1	15	++	+++	/	/	---	---	Distal UL postural tremor
5	*LITAF*	c.404C > G p.Pro135Arg	5.1 (p)	41	+	++	/	++	---	---	Distal UL postural tremor
			5.2	6	/	+	/	/	/	--	N.A.
6	*GARS1*	c.1660G > A p.Asp554Asn	6.1 (p)	53	/	+	/	/	--	/	N.A.
			6.2	28	/	/	/	/	/	/	N.A.
			6.3	14	/	/	/	/	/	/	N.A.
7	*GDAP1*	c.374G > A p.Arg125Gln	7.1	52	/	+++	/	/	+++	++	Proximal LL weakness, pyramidal signs
8	*GDAP1*	c.374G > A p.Arg125Gln	8.1	50	/	++	/	++	+++	/	Asymmetric LL weakness, optic nerve subatrophy, neurosensorial hypoacusia, pyramidal signs

yrs: years; (p): proband; UL: upper limbs; LL: lower limbs; +: mild; ++: moderate; +++: severe (for reflexes only: +, ++, +++: mild, moderate and great increase; --, ---: moderate reduction and absence); /: normal; N.A.: not applicable.

**Table 4 brainsci-11-01616-t004:** Neurophysiological studies of 12/17 subjects.

Family	Gene	Variant	Pt.	Neurophysiologic Phenotype	Ulnar Motor			Ulnar Sensitive			Peroneal			Sural	
					Amplitude	Velocity	Onset Latency	Amplitude	Velocity	Onset Latency	Amplitude	Velocity	Onset Latency	Amplitude	Velocity
1	*BSCL2*	c.263A > G p.Asn88Ser	1.1 (p)	Axonal	10	57	2.4	48	61	2	0	0	0	8	61
			1.2	Axonal	12	69	2.3	79	71	2.1	2	41	6	28	46
			1.3	Axonal	14	57	2.2	59	86	2	3	46	6.2	6	40
			1.4	Axonal	14	55	2.8	44	60	5.1	1	39	5.2	27	59
			1.5	/											
2	*BSCL2*	c.263A > G p.Asn88Ser	2.1 (p)	Axonal	18	58	2.6	93	63	2.2	1	40	4	12	45
			2.2	Axonal	6	51	3.4	20	50	3	0	0	0	11	44
			2.3	/											
3	*MORC2*	c.1503A > T p.Gln501His	3.1	Intermediate	13	33	5.3	10	36	5.5	3	22	9.7	0	0
4	*HINT1*	c.110G > C p.Arg37Pro	4.1	Intermediate	10	45	4.1	35	55	2.1	1	34	7.7	5	33
5	*LITAF*	c.404C > G p.Pro135Arg	5.1 (p)	Demyelinating	9	31	6.8	12	37	5.3	6	22	8.6	0	0
			5.2	/											
6	*GARS1*	c.1660G > A p.Asp554Asn	6.1 (p)	Axonal	5	52	4.9	24	65	3.5	0	0	0	12	45
			6.2	/											
			6.3	/											
7	*GDAP1*	c.374G > A p.Arg125Gln	7.1	Axonal	3	39	4.8	48	61	2	0	0	0	8	61
8	*GDAP1*	c.374G > A p.Arg125Gln	8.1	Axonal	9	48	4.1	44	60	5.1	0	0	0	27	59

Pt.: patient; (p): proband;/: not performed.

## Data Availability

The data presented in this study are available on request from the corresponding author.

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
