# Peer review of "Rare among Rare: Phenotypes of Uncommon CMT Genotypes"

_brainsci, 2021, doi:10.3390/brainsci11121616_

Round 1

Reviewer 1 Report

Please consider italics for human gene names according to convention / publisher style. 

MDPI | Layout Style Guide

Ln 52: From eight unrelated families

Ln 54-55: Gene selection is of utmost importance in this kind of study. Please include the information in an appendix table. Was it based on an exome kit or was it a custom panel? See my comment on Results as well. 

Ln 60: I have experience with InterVar and can say that while it is open to the academic community and very convenient for batch jobs, its functionalities still lags behind some commercial solutions. I would suggest the authors at least verify suspected pathogenic variants (even though batch submission may require paid subscription) with an alternative commercial engine, e.g. VarSome The Human Genomics Community This is to make sure enough in silico analyses have been properly done.

Ln 61-62: For simple audit studies, ethics approval may be waived... but this depends on institutional policy. Please state the IRB approval number (if available), or state it was not needed, to avoid doubt. 

Table 2: The presentation in the Table is strange. First column should be re-formatted such that the extra row is removed, the first row / first cell becomes 1 & 2 or 1, 2.; and last cell becomes 7 & 8 or 7, 8 It looks strange to separate otherwise identical information into two rows. Some transcripts in the Table lack a version number. 

Results: I do not have major comment about the Results section. It is the whole point of the study anyway. 

One particular view I have about the presentation is that the authors should avoid stating the unusual phenotypes as CMT-related at this stage. A more conservative wording could be fine, e.g. "atypical clinical features" could be better than "special clinical features" (Table 1); Table 3, as a result, is fine. My reasoning is that if the 44-gene panel is all that has been done, patients could well have other pathogenic variants in other genes that can explain the atypical findings - and attributing high CSF protein, pigmentosus retinits or cataract to a CMT-gene (without further evidence) could be problematic. 

Conclusions: The comment that NGS / WES are still not widespread should be qualified. They may be less available in some health systems but both are rapidly becoming the norm in developed countries. 

Reviewer 2 Report

Dear authors,

thank you for intereseting manuscript, some suggestions for improvement.

1) gene names should be written in italic in some places it is is in some it is missing.

2) according to HGVS term - mutation usage should be avoided but replaced by disease causative variant or pathogenic/ likely pathogenic variant

3) also in abstract suggested to replace next generation sequencing with more specific technique used - exome sequencing, gene panel or genome sequencing.

4) in methods section InterVar is used - is it also using Phenotype data - as it is important pathogenicity criteria?

5) when in results - it is divided by the genes - there would be recommended OMIM numbers or HGNC ID used as well reference sequence because it will allow avoiding of incorrect causative variant nomenclature usage.

6) for methods - either list of literature or other explanation would be needed what was the basis of gene list selection.

7) also change in result would be suggested in the first general description of the genes and then started one by one in detail, in another case, it is a bit unclear why starting with one gene there is given a table where is shown all unusual genes?

8) for table 1 also un the name should be given references, not only in the text. or clinical features are observed in this study? suggested also o write that those phenotypes are related with CMT

9) causative variants should be written in one way if translated in protein level either aminoacuds are abbreviated as Arg or used single letter as R, e.g. for the BSCL2 variants in te text are p.S90W, and N88T even without "p." and in the table p.Asn88Ser

10) what is shown in the Table2 variants from the other reports? it mixed, as table should be understandable alone without text that is not clear. Polyphen and Sift are only two of may other in silicon pathogenicity tools, so I would suggest deleting both columns also cosegregation, and Gnomad frequency, but substituting with fulfilled ACMG criteria not only the result of the variant class and probably if it is mentioned in ClinVar and the allelic ID.

11) table titles should be changed, to re[resent the content more, that table could be read also without the main text. 

12) for families - pedigrees would be suggested for better illustration.

13) results are described and also discussed - so would suggest also to show in the title - results and discussion, or the results from the other studies should be moved to discussion.

14) variant in the GARS  gene is ClinVar database is reported as pathogenic https://www.ncbi.nlm.nih.gov/clinvar/RCV001542257/ in the manuscript as VUS - what is the reason for the discrepancy?

Round 2

Reviewer 1 Report

The many amendments - including some typos in gene names the reviewer missed - were noted. When diseases involve rare genes, it is worthwhile to double-check the information before submission. 

Some language polishing could make the manuscript more fluent, otherwise this is a much improved version. Congratulations!

Author Response

Thank you for your revision, comments, suggestions and also for your congratulations!!

Best regards